# What Else Can AI See in a Digital ECG?

**DOI:** 10.3390/jpm13071059

**Published:** 2023-06-28

**Authors:** Tomasz Rechciński

**Affiliations:** Chair and Department of Cardiology, Medical University of Lodz, 91-347 Lodz, Poland; tomasz.rechcinski@office365.umed.pl; Tel.: +48-42-251-61-12

**Keywords:** ECG, artificial intelligence, prediction, cardiovascular events, progression of cardiovascular disease

## Abstract

The electrocardiogram (ECG), considered by some diagnosticians of cardiovascular diseases to be a slightly anachronistic tool, has acquired a completely new face and importance thanks to its three modern features: the digital form of recording, its very frequent use, and the possibility of processing thousands of records by artificial intelligence (AI). In this review of the literature on this subject from the first 3 months of 2023, the use of many types of software for extracting new information from the ECG is described. These include, among others, natural language processing, backpropagation neural network and convolutional neural network. AI tools of this type allow physicians to achieve high accuracy not only in ECG-based predictions of the patient’s age or sex but also of the abnormal structure of heart valves, abnormal electrical activity of the atria, distorted immune response after transplantation, good response to resynchronization therapy and an increased risk of sudden cardiac death. The attractiveness of the presented results lies in the simplicity of the examination by the staff, relatively low costs and even the possibility of performing the examination remotely. The twelve studies presented here are just a fraction of the novelties that the current year will bring.

## 1. Introduction

The recording of the electromagnetic activity of the heart using electrodes placed on the surface of the human body has been known in the diagnosis of heart diseases for nearly 120 years [1,2,3]. Analogue methods of the recording of changes in the vector of electromagnetic forces have been replaced with digital recording over time, and the interpretation of changes in the ECG curve, initially based on the assessment of P, Q, R, S and T waves by means of the human eye, has been replaced with automatic analysis [4,5]. Collecting thousands of digital ECG records in databases and confronting them with diagnostic methods based on cardiac imaging as well as with clinical data allows us to look at the ECG curve as an inexhaustible source of information about patients [6,7,8]. This happened thanks to artificial intelligence (AI), a tool which saw much more in the ECG than the human eye [9,10].

## 2. Methods

This article is a review of scientific reports from the open access group published in the first 3 months of this year and found in the PUBMED database after entering the two abbreviations used in the title of this work. After the rejection of case reports, papers on imaging or on IT (information technology) methodology, descriptions of projects and review papers from the 31 publications, there remained 12 original papers, which I would like to arrange thematically and present—Table 1.

## 3. Results

A group of researchers from the Mount Sinai hospital network in the US decided to investigate the usefulness of AI in ECG screening for the early detection of common valvular heart diseases: aortic stenosis and mitral regurgitation. With approximately 617,000 digital ECGs retrospectively paired with cardiac ultrasounds from approximately 123,000 patients from five collaborating centers (including 23,484 mitral regurgitation patients and 9399 aortic stenosis patients), the natural language processing stream (an informatics tool known for three decades) was developed to capture specific terms from echocardiogram (ECHO) descriptions, related to the valvular defects listed above [23,24]. The most interesting result of this study was the finding of the high accuracy of ECG assessment by AI for detecting aortic stenosis before ECHO was performed. For this heart defect, it was found that the performance of the model increased inversely with the time interval between the ECG and the first ECHO. In the period of 18–24 months before the first ECHO, the area under the ROC curve was 0.66, and in the period of 6–12 months and 3–6 months it increased to 0.72. Using the Youden index, the created model achieved a sensitivity of 0.84, 0.87 and 0.95 for the above-mentioned periods of time preceding the ECHO which determined the diagnosis of this heart defect. For mitral regurgitation, which in the studied cohorts of patients was 3–4 times more common than the other heart defect, the sensitivity index for the ECG analyzed by AI was 0.94 for the ECHO performed in the hospital and 0.83 for the ECHO performed in an outpatient setting, with a specificity of 0.69 and 0.63, respectively. The authors conclude that this inexpensive and widely available diagnostic tool, i.e., the ECG, when analyzed by AI, can make the diagnosis of aortic stenosis or mitral regurgitation earlier and thus significantly improve the patient’s prognosis thanks to the earlier implementation of treatment. The authors point out some limitations of their study. Firstly, while the developed model works well in the long-term development of the described heart defects through gradual changes in the ECG signal in these patients, it may fail in the case of acute ischemic mitral regurgitation because the heart muscle, and thus the ECG signal in these patients, will not be able to adapt to the new conditions. Secondly, the different incidence of heart defects in inpatients and outpatients was also underlined, which affected the value of statistical parameters (positive predictive value). The researchers encourage other cardiology centers to try their model [11].

In another study, a group of physicians and computer scientists from Taiwan used traditional electrocardiographic criteria for the diagnosis of left ventricular hypertrophy (R and S wave amplitude) and then the backpropagation neural network (BPN) method to improve the accuracy of the diagnosis of left ventricular hypertrophy confirmed by ECHO as a method that is the gold standard. Between 8 and 13 evolutions of 1466 ECG records, which in the case of 173 people came from patients with echocardiographically confirmed left ventricular hypertrophy, were the material for creating a model which detected left ventricular hypertrophy on the basis of a digital ECG. This model achieved the accuracy, precision, sensitivity and specificity of the test at the level of 0.961, 0.958, 0.966 and 0.956, respectively, which significantly exceeds the results obtained using the traditional methods of visual ECG assessment. Some limitations of this work should be emphasized. All the ECGs from this study were obtained from subjects without arrhythmias (e.g., atrial fibrillation) and with heart rates between 60 and 100/min. The authors are making plans which will make it possible to adapt this model for a wider range of heart rates, including patients with arrhythmias. In the future, AI may help cardiologists to analyze a large number of ECG records, especially when ECGs of hundreds of apparently healthy people need to be screened on a massive scale. When assessing the future prospects of this method, it would be valuable to be able to stratify the risk of diastolic heart failure or the risk of sudden cardiac death or of acute myocardial infarction in people with confirmed myocardial hypertrophy [12].

AI prediction of the most common arrhythmia, which is atrial fibrillation (AF), on the basis of the ECGs of patients with sinus rhythm, has been the subject of previous studies, but recently a group of researchers from the USA decided to check the usefulness of ECGs obtained using mobile devices (mECGs) operated by the patients themselves outside healthcare facilities. The study included 73,861 Alivecor KardiaMobile 6L users (mean age 58.14 years; 65% male) and delivered 267,614 mECGs. From this number of mECGs, 60.15% of the reports came from users with paroxysmal AF. The authors tested the study model on mECGs’ sinus rhythm recordings 0–2 days before and after, 3–7 days before and after, and 8–30 days before and after an AF attack to determine the optimal screening window and finally tested the model on pre-onset mECGs to determine whether AF can be predicted prospectively. The model’s performance on the mECG test set from patients with and without AF on all windows of interest showed an area under the curve (AUC) value of 0.760 (95% confidence interval [CI] 0.759–0.760), sensitivity of 0.703 (95% CI 0.700–0.705), specificity of 0.684 (95% CI 0.678–0.685) and accuracy of 69.4% (95% CI 0.692–0.700). Model performance was best for the mECGs from 0–2 days before or 0–2 days after the AF paroxysm (sensitivity 0.711; 95% CI 0.709–0.713) and worse in the window before and after 8–30 days (sensitivity 0.688; 95% CI 0.685–0.690). As was shown, neural networks are able to predict AF using widely scalable and cost-effective mobile technology, both prospectively and retrospectively. When discussing the methodological limitations of this study, the authors point out the necessity of a certain compromise between the good quality of the ECG recording and the ease of use of the device when performing AI-based analysis using mECG. They emphasize that the Alivecor AF detection package does not distinguish between atrial fibrillation and atrial flutter. In addition, those results only included participants using the Alivecor 6L device and may not be “extrapolated” to users of other mECG devices on the market. They also consider the unavailability of information such as clinical indicators, socio-economic status or racial background to be an important limitation which limits the use of the presented method in everyday medical practice [13].

Interesting results of studies on the prediction of AF recurrence after catheter ablation of the pulmonary veins were presented by researchers from Guangzhou, China. The study enrolled 1618 patients aged 18 years and over with paroxysmal atrial fibrillation (pAF) who underwent pulmonary vein isolation (PVI) by experienced operators. A 12-lead ECG taken within 30 days before PVI was used by the convolutional neural network (CNN) to predict the risk of arrhythmia recurrence. Compared to the current prognostic models, such as APPLE, BASE-AF2, CAAP-AF, DR-FLASH and MB-LATER, the performance of the AI algorithm created by the authors for predicting AF recurrence was better [25,26,27,28,29]. The predictive performance of the AI-enabled ECG was assessed on the basis of the area under the curve (AUC). The AUC of the AI algorithm in the ECG analysis for predicting FA recurrence after ablation was 0.84 (95% CI: 0.78–0.89), and the sensitivity, specificity, accuracy and precision were 72.3%, 95.0%, 92.0% and 69.1%, respectively. An effective method for predicting the risk of recurrence in patients with pAF after PVI is of great importance when deciding on personalized ablation strategies and postoperative treatment prospects. However, in that study, the sample size was small, which the authors considered to be one of the main limitations. They note that a prospective study will be needed to confirm the predictive validity of that model. Moreover, the proposed model could predict the risk of recurrence only 3–12 months after ablation [14].

A group of researchers from Tokyo hospitals assumed that the previously used predictive models of sudden cardiac death (SCD) in patients with heart failure (HF) did not give satisfactory results. They assessed the usefulness of electrocardiography-based AI (ECG-AI) in better predicting SCD and tested whether combining the ECG-AI index and conventional SCD predictors would improve SCD stratification in HF patients. The prospective observational study included 2559 patients hospitalized for HF who were discharged home after acute decompensation. The ECG data from the hospitalization period came from electronic medical records systems. The relationship between the ECG-AI index and SCD was assessed by taking into account left ventricular ejection fraction (LVEF), New York Heart Association (NYHA) class and the competing risk of non-SCD. The ECG-AI index combined with classical predictors (LVEF ≤ 35%, NYHA class II and III) resulted in a significant improvement in the discriminant value for SCD. The Fine–Gray model used by those authors, taking into account the competing risk of non-SCD, showed that the ECG-AI index was independently associated with SCD (HR 1.25; 95% CI, 1.04–1.49; *p* = 0.015). The researchers observed also an increased proportional risk of SCD vs. non-SCD along with an increase in the ECG-AI index (low, 16.7%; intermediate, 18.5%; high, 28.7%; p for the trend = 0.023). This study concludes that for the improvement of SCD risk stratification, the use of ECG-based AI may provide added value in the treatment and primary prevention for patients with HF. In addition to the statistical limitations resulting from the small number of patients in general and those with dilated or hypertrophic cardiomyopathy in particular, the authors pay attention to the racial aspect—only East Asians participated in the analysis; therefore, the usefulness of the results of this study for other regions and races would have to be confirmed in subsequent studies. They emphasize that at the time of data collection, patients with heart failure in Japan did not have access to sacubitril-valsartan and flozin—drugs that reduced mortality in this group of patients. The researchers are also somewhat critical and cautious about the use of AI in medicine, concluding in their discussion that the pitfall of AI models is that the dataset can contain errors that may lead to misclassification or other unidentified errors [15].

Another team of Japanese doctors have noted that there is growing evidence that the 12-lead ECG can be used to predict biological age, the value of which is related to the rate of cardiovascular events. On the other hand, the usefulness of age predicted by AI using ECG remains unexplored. Those researchers made use of the single-center Shinken Database belonging to the Cardiovascular Institute in Tokyo and developed an AI-enabled ECG to evaluate 17,042 ECGs with sinus rhythm (SR-ECG) in order to predict chronological age (CA) using a convolutional neural network which allows you to determine the age predicted by artificial intelligence. The study included all patients with sinus rhythm who underwent cardiovascular examinations in 2010–2018, excluding foreigners, people under 20 and over 90 years of age and people with active cancer. The AgeDiff parameter was defined as the difference between the age predicted by AI and the chronological age, and then three categories were distinguished: AgeDiff < −6, from −6 to ≤6 and >6 years. During the mean follow-up of 460.1 days, there were 543 cardiovascular events. The incidence of cardiovascular events over 12 months was 2.24%, 2.44% and 3.01%/year for patients with AgeDiff < −6, from −6 to ≤6 and >6 years, respectively. The AUCs for cardiovascular events predicted by CA and age given by AI on the basis of ECG were 0.673 and 0.679 (the Delong test, *p* = 0.388) for all patients; 0.642 and 0.700 (*p* = 0.003) for patients with chronological age < 60 years); and 0.584 and 0.570 (*p* = 0.268) for those with chronological age ≥ 60 years, respectively. Hence, the age predicted by AI on the basis of a 12-lead ECG showed an advantage in predicting cardiovascular events compared to CA, but only in patients < 60 years of age. Among the limitations, the authors of the publication mention the fact that their study was a single-center study, so the results would require confirmation in groups of patients from other hospitals or people from the general population. One of the inclusion criteria was ECG sinus rhythm, so the conclusions cannot be extrapolated to patients with arrhythmias. Important patient data such as cardiac morphology, comorbidities including frailty syndrome, or the effects of medication were not included in the model [16].

The study by another group—cardiologists, internists and endocrinologists from Mayo Clinic in Rochester, Minnesota, USA—confirmed that the ECG curve reveals not only age but also sex. The AI-ECG was used to determine the likelihood of being male in a study where ECGs collected over 21 years (2000–2020) were paired with the results of blood tests for sex hormones: total testosterone and estradiol. Total testosterone and ECG were used in 58,084 male and 11,190 female patients, and estradiol and ECG in 2835 male and 18,228 female patients. Total testosterone concentrations showed a moderate positive correlation (r = 0.46) with AI-ECG male probability—*p* < 0.001, which was quantified in the range of 0.0 to 1.0, with higher AI-ECG values indicating a high probability of being male. On the other hand, in women with a high probability of male sex according to AI-ECG, statistically significantly higher concentrations of total testosterone and lower concentrations of estradiol were found in comparison with persons with a low probability of male sex according to AI-ECG, regardless of the age of the subjects. This study showed that the AI-ECG can be used as a biomarker of endocrine status in terms of sex hormones [17].

The use of AI-ECG has also aroused the interest of transplantologists—I would like to present two recently published papers on this subject. The first, from Mayo Clinic, concerns the possibility of predicting the risk of rejection of a transplanted heart on the basis of AI-ECG, and its results may contribute to an improvement in the care of patients after heart transplantation by providing not only a quick and non-invasive method but also a remotely usable screening method to predict the recipient’s response to a transplanted heart. The heart transplant recipients came from three Mayo Clinic centers, where they were treated between 1998 and 2021. Digital 12-lead ECGs and endomyocardial biopsy results were extracted from their medical records. The researchers used the criteria of the International Society for Heart and Lung Transplantation for moderate to severe acute cellular rejection of an allograft. They collected data from 1429 patients, for whom 7590 ECG-biopsy pairs were available. Retrospectively, the AI-ECG detected acute cellular rejection of the heart transplant with the AUC of 0.84 [95% (CI): 0.78–0.90] and a sensitivity of 95% (19/20; 95% CI: 75–100%). Also, a prospective screening study confirmed the validity of the concept from the authors of this study (56 patients; 97 ECG-biopsy pairs)—AUC = 0.78 (95% CI: 0.61–0.96) and 100% sensitivity (2/2; 95% CI: 16–100%). What is most important, after this somewhat limited study, is to make it clear that a positive AI-ECG-based screening result is not intended to dictate therapeutic intervention but rather to support the need for confirmatory testing by endomyocardial biopsy. And a negative AI-ECG screening result would be reassuring and, in the absence of other clinical evidence of rejection, would allow the biopsy to be postponed at the discretion of the attending physician [18].

The second paper in the field of transplantology deals with the preoperative stratification of the risk of left ventricular dysfunction or of perioperative atrial fibrillation in patients requiring liver transplantation. Like the previous one, this study comes from Mayo Clinic in the US. The methodology used pre-trained AI which aimed to detect the possibility of left ventricular systolic dysfunction (LVEF < 50%) or of periprocedural atrial fibrillation in a standard 12-lead digital ECG. The researchers analyzed 3593 ECGs and 1110 ECHOs both from patients assessed immediately before transplantation and from patients assessed retrospectively from the databases from previous years—a total of 712 people. On the basis of the results, which showed 96.1% sensitivity and 52.3% specificity of the AI-ECG analysis in predicting AF, and the probability of 0.2 or greater that patients with EF < 50% will experience a reduction in LVEF, this conclusion was made: the use of the AI-ECG algorithm is a low-cost option to obtain important information on cardiac risk, and it is easy to implement before qualifying a patient for a liver transplant [19].

Researchers from Maastricht, Utrecht and Groningen in the Netherlands have developed a new deep learning algorithm which can predict the effect of cardiac resynchronization therapy. It was called the FactorECG, and its ability to predict the response to resynchronization therapy and the occurrence of the so-called endpoints, i.e., death, the need for an implantation of a left ventricular assist device or the need for heart transplantation, was compared with the methods used so far: AHA guidelines, ESC guidelines, calculating the area under the QRS curve and prediction based on clinical data. The deep learning algorithm was trained on 1.1 million ECG records from 251,473 patients. After obtaining the averaged ECG records, the features of each ECG were summarized in the 21 variables that make up the FactorECG. The FactorECG was created for the pre-operative ECGs from 1306 patients undergoing cardiac resynchronization therapy. Although the FactorECG predicted the occurrence of endpoints better than the guidelines of scientific societies and the area under the QRS curve, the best results were obtained after strengthening the prediction with clinical data: the c-statistic with a 95% confidence interval was 0.69 (0.66–0.72) for the FactorECG itself and 0.72 (0.69–0.75) for the FactorECG enhanced with clinical data. When the FactorECG was compared with the area under the QRS curve for predicting response to resynchronization therapy, its c-statistic matched the calculation of the area under the QRS curve only after it was enhanced with clinical data [0.7 (0.67–0.74) for both methods]. In the conclusions, the authors state that the FactorECG facilitates personalized decision making in qualifying patients for resynchronization therapy, and at the same time, it is easy to use, which enables its efficient use in everyday practice. The authors emphasize that the advantage and strength of their study is that it was conducted on a very large number of ECG records. Although they came from one supplier of ECG equipment, as the authors of this publication emphasize, previous studies had shown that the results of deep learning based on ECG generalize well to other groups where studies were conducted on the basis of equipment from other manufacturers of ECG devices. There are also some limitations to this study, notably that the ethnicity of the patients and the cause of death were not included in the database and that the results presented cannot be generalized to patients receiving an update to the cardiac resynchronization therapy [20].

Authors from Portugal can talk about the high specificity of their method of diagnosing pulmonary embolism using AI-ECG. They compared the effectiveness of their method with traditional scales for assessing the probability of pulmonary embolism—the Wells score and the Geneva score. Nine hundred and eleven ECGs from patients admitted to the emergency department for chest CT scans because of suspected pulmonary embolism were used to develop the AI model, and one hundred and three ECGs from such patients were used for validation. The AI-ECG model achieved 100% specificity with the confidence interval (CI) of 94–100 and 50% sensitivity (CI 33–67) in detecting pulmonary embolism. Also, this study provides evidence to support the use of artificial intelligence in routine clinical practice, which will improve the accuracy of management of patients with this life-threatening disease. As the last of the studies presented in this review, it differs in some ways in its approach to the use of AI in cardiac diagnostics from the previous studies. The differences are that the ECG was the only test needed to determine the risk of pulmonary embolism; no other clinical or laboratory data were required. Moreover, patients with arrhythmias were not excluded from this study, which means that the model used was applicable to patients with AF or with an artificial cardiac pacemaker. Finally, patients were not categorized according to the severity of the clinical condition, the model determined the risk of both low-risk and high-risk embolism with imminent death. The authors emphasize that their work is pioneering and will require confirmation by other centers where patients with pulmonary embolism are admitted [21].

At the end of this review, I would like to present a work that shows that AI-ECG can be used not only in medicine but also in identity verification. A study on this subject was conducted by researchers from Canada. Using the database of the Physikalisch-Technische Bundesanstalt (PTB), they showed that ECG data can be collected from users to create a unique biometric profile for each person. ECG has proven to be one of the most reliable advanced authentication techniques; unlike other biometric methods (fingerprints, facial image, eyes, palm veins, speech or the shape of the ears), it confirms that a person is real and alive. Perhaps in the future, this approach will provide secure and convenient user authentication in various situations that require identity verification. In order to make ECG-based identification more secure, several strategies should be adopted today. These include the need to provide better quality ECG recordings, the need to develop more advanced deep learning models that can extract significant features from ECG data, the need to adopt secure data transmission protocols such as encryption and SSL (Secure Socket Layers), which can help protect the transmitted data, and the need to implement multi-factor authentication, which requires multiple forms of identification in order to access the system (apart from the ECG, the system should require passwords or security tokens, facial image, fingerprints, etc.). It would be extremely important to regularly update and test the ECG-based deep learning authentication system to ensure its effective operation and satisfactory security [22].

## 4. Conclusions

The question posed in the title is still open; certainly, the next months of 2023 will bring more publications on new and surprising applications of AI-ECG. Their number is probably unlimited, just as human ingenuity is infinite [30].

## Figures and Tables

**Table 1 jpm-13-01059-t001:** The content of the presented review, topics of the selected publications and corresponding reference numbers.

Subject of Interest	Reference Number
Valvular diseases	[11]
Left ventricular hypertrophy	[12]
Onset of atrial fibrillation	[13,14]
Risk of sudden cardiac death	[15]
Biological age of the patient	[16]
Concentration of sex hormones	[17]
Heart transplant rejection	[18]
Cardiovascular complications of liver transplantation	[19]
Effects of cardiac resynchronization therapy	[20]
Pulmonary embolism	[21]
Identity verification	[22]

## Data Availability

The data presented in this study are openly available in the PUBMED database.

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
