# Peer review of "What Else Can AI See in a Digital ECG?"

_jpm, 2023, doi:10.3390/jpm13071059_

Round 1
Reviewer 1 Report
Dear Professor Rechciński,
I am writing to recommend the publication submitted to JPM. After a thorough review, I am convinced that this article is a valuable contribution to the field of cardiology. The research methodology is sound and the results are well-organized and clearly presented.
However, I would like to suggest that the English language used in the manuscript could be improved. There are several areas where the language could be refined to enhance the clarity and readability of the text. For example, there are a few instances where the grammar could be improved or where the word choice could be more precise.
Nonetheless, these issues are relatively minor and can easily be addressed with revisions. Overall, the quality of the research and the significance of the findings justify publication of this article in JPM.
Therefore, I recommend that this manuscript be accepted for publication after minor revisions. I am confident that with the suggested improvements, this article will make an important contribution to the field and be well-received by readers.
Thank you for your consideration of this recommendation.
Sincerely,
I think its worth publication after some minor revisions/ english improvement.
Author Response
Dear Reviewer, Thank you for your suggestion.
The manuscript was linguistically consulted twice, both before and after peer review. Both consultants,
who are professional translators of medical texts, stated: "To me, the text looks perfectly clear and
readable" and " I am of course ready and willing to improve on the text and to enhance its clarity and
readability, provided, however, that the Reviewer points out specific examples of vocabulary, grammar
structures or other instances of usage which offend his or her linguistic sensitivity (Jane Austen would
have written “sensibility”)".
So I would like to ask Reviewer 1 to point out specific language defects that he or she does not like
Reviewer 2 Report
I don't understand what the scholarly contribution of this article is. Just excerpting and compiling 12 abstracts and conclusions? The reader can themselves quickly filter what is of interest from the titles and abstracts and exclude what they don't want to read from the article types by searching for the time period, even if the numbers are thirty.
Articles that can be called Reviews have rigorous requirements. Neither the comprehensive organization nor the expert summarization can be found in this manuscript.
Author Response
Dear Reviewer, Thank you for your suggestions.
I do not share the opinion of Reviewer 2 regarding the redundant nature of the work submitted for publication.
Reviewer 3 Report
The paper “What else can AI see in a digital ECG?” is an interesting review about a hot topic argument: AI applied to medicine.
Twelve very recently papers about AI applied to ECG have been summarized and commented; very different chances concerned the availability of AI applied to ECG to discriminate: valvular heart diseases, left ventricular hypertrophy, atrial fibrillation, sudden cardiac death risk, left ventricular dysfunction after heart transplant, response to CRT, pulmonary embolism, age and gender and also ECG used as identity verification.
The Authors described very well all the papers, with an open mind to future application of AI to ECG and others tools.
The last sentence (about human ingenuity) is very appropriate but may be too personalized
Only a clarification: at line 41 Authors should specify what does IT mean.
Author Response
Dear Reviewer,
Thank you for your suggestions.
An explanation of the abbreviation IT – "information
technology" – has been added.
Reviewer 4 Report
The author provides a review of the AI applications in ECG from the first 3 months of 2023. General concept comments: The focus of the review article on just a brief period of 3 months is odd. The author needs to use a wider time for the review or provide a detailed justification for why this extremely abbreviated time period was chosen. Besides the shorter review period of 3 months, the article must be improved by providing the following: • Summary of related review works in AI-ECG • Review questions and rationale for synthesizing the empirical evidence for this systematic research work • Methodology employed for reviewing the literature (for example, PRISMA methodology) • Tabular summary of review articles I recommend re-submitting the article after the comments are satisfactorily addressed.
Author Response
Dear Reviewer,Thank you for your suggestions.
The short period of time from which the publications come was
dictated by both the already available review papers regarding publications from 2022 and earlier, and,
on the other hand, the deadline for submitting papers for the special issue. Despite the short period of
time, I managed to collect interesting material, which is related to the wealth of publications on artificial
intelligence and ECG.
The purpose of this work was not to present my own opinion on the selected topic, nor to comment on
the presented works. A tabular summary of the manuscript has been added.
The methodology for selecting the papers described after the introduction is very similar to that
recommended by the PRISMA guidelines; however, this work does not pretend to be a comprehensive
systematic review, but simply a review
Round 2
Reviewer 2 Report
I feel that the author's response does not address the main concerns raised by the reviewers.
Here I point out furthermore a methodological problem and an issue arising from the responses:
1. "entering the two abbreviations used in the title of this work." i.e., AI and ECG. I doubt that's right and comprehensive. For example, some new results appeared when I replaced ECG with electrocardiogram or electrocardiography.
2. I disagree with "The short period of time from which the publications come (which was the complaint of Reviewer 4) was dictated by both the already available review papers regarding publications from 2022 and earlier..." in the author's response. Just one example:
Let's go forward a month. As far as I know, a novel method published at MDPI in December 2022, https://doi.org/10.3390/bios12121182, was cited nearly 10 times within four months. Applying a purely statistical model instead of training, this AI method can efficiently and conveniently find outliers in the ECG, automatically segment ECG signals without supervision, and facilitate pulsus paradoxus analysis, etc.
It fits the title "What else can AI see in a digital ECG" perfectly and supplement the topics in Table 1.
From the above example, it is also perceivable that the author’s search keywords may miss some valuable articles. They do contribute to AI-based ECG research, but they do not use both ECG and AI as keywords. They can be BIOsignal, BIOMEDICAL signal, PHYSIOLOGICAL signal, or VITAL sign signal...
If this is a comprehensive review of a subject topic with numerous references, I think it is reasonable not to widen the search. However, an article that covers only a very short period and a narrow topic should take more comprehensive measures to reflect its academic value better.
I do not deny this manuscript's contribution, but I think the current compiling is not comprehensive to be a review.
Author Response
I
am not in favor of changing the inclusion and exclusion criteria after obtaining the results, whatever
they may be. The choice of the first three months of this year - is certainly an arbitrary choice, I take
the entire burden of such a choice on myself and sign it with my name.
Reviewer 4 Report
The prior reviewer's comments on the short period of time from which the publications come is not properly addressed in this revision. While the author's constraint on the deadline for submitting papers for the special issue is understandable, a scientific justification for the shorter review period (first 3 months of 2023) is not provided. Given the broad scope of the article title, it is recommended that the author provide a detailed review on the topic over a longer time period.
Author Response
Of the selected references, 12 are selected in accordance with the methodology described in
the work, and the remaining ones concern the history of the electrocardiogram or research to which
the authors of works on the use of AI in ECG analysis refer
I am not in favor of changing the inclusion and exclusion criteria after obtaining the results, whatever
they may be. The choice of the first three months of this year - is certainly an arbitrary choice, I take
the entire burden of such a choice on myself and sign it with my name.